# Risk Factors for Non-Healing Wounds—A Single-Centre Study

**DOI:** 10.3390/jcm13041003

**Published:** 2024-02-09

**Authors:** Daniel Wolny, Ladislav Štěpánek, Dagmar Horáková, Janet Thomas, Jana Zapletalová, Mihir Sanjay Patel

**Affiliations:** 1Department of Public Health, Faculty of Medicine and Dentistry, Palacký University Olomouc, Hněvotínská 976/3, 775 15 Olomouc, Czech Republic; daniel.wolny01@upol.cz (D.W.); dagmar.horakova@upol.cz (D.H.); janet.thomas01@upol.cz (J.T.); mihirsanjay.patel01@upol.cz (M.S.P.); 2Department of Surgery, Military Hospital Olomouc, Sušilovo Náměstí 5, 771 11 Olomouc, Czech Republic; 3Department of Biophysics, Faculty of Medicine and Dentistry, Palacký University Olomouc, Hněvotínská 976/3, 775 15 Olomouc, Czech Republic; jana.zapletalova@upol.cz

**Keywords:** risk factor, healing process, wound, diabetic ulcer, pressure ulcer, vascular ulcer

## Abstract

**Background:** Chronic wounds present a significant clinical, social, and economic challenge. This study aimed to objectify the risk factors of healing outcomes and the duration of chronic wounds from various etiologies. **Methods:** Patients treated for non-healing wounds at the surgical outpatient clinic of the Olomouc Military Hospital were involved. Data from patients treated between 8/2021 and 9/2023 were selected. Patients were mostly treated as outpatients, with microbiological follow-up indicated in cases of advanced signs of inflammation. **Results:** There were 149 patients who met our selection criteria (the mean age was 64.4 years). Predominant causes of wounds involved diabetes (30.9%), post-trauma (25.5%), pressure ulcers (14.8%), surgical site infections (14.8%), and vascular ulcers (14.1%). Patient outcomes included wound resolution in 77.2% of patients (with a mean healing time of 110.9 days), amputation in 14.1%, and wound-related death in 8.7% of patients. Non-healing cases (amputation/death) were predicted by several local factors including an initial depth greater than 1 cm, wound secretion, inflammatory base, and a maximum wound size. Systemic factors included most strongly clinically manifested atherosclerosis and its risk factors. Of the 110 swabs performed, 103 identified at least 1 bacterial genus. The dominant risk factor for a prolonged healing duration was bacterial infection. Wounds contaminated by *Proteus* or *Pseudomonas* had prolonged healing times of 87 days (*p* = 0.02) and 72 days (*p* = 0.045), respectively. **Conclusions:** The early identification of local and systemic risk factors contributes to the successful resolution of chronic wounds and a reduced duration of healing.

## 1. Introduction

Wound care is an essential component of surgery. Given the breadth of the field of surgery, it may seem that wound care is only a marginal component of surgical treatment. However, given the ageing population, an increasing burden of diabetes and obesity worldwide, and the persistent risk of infection, it is expected that chronic wounds will continue to pose a substantial clinical, social, and economic challenge in surgical care [1,2,3]. Chronic wounds are those that do not progress through a normal, orderly, and timely sequence of resolution. Common wounds of the lower extremities include arterial, diabetic, pressure, and venous ulcers. The prevalence of chronic wounds has been compared to the prevalence of heart failure, with the associated mortality rate of failed healing exceeding that of some cancers [4]. The morbidity and associated costs of chronic wounds highlight the need to improve wound prevention and treatment guidelines [5].

Effective management should rely on a standardized approach to the healing of chronic wounds. Currently, this approach is absent in the Czech Republic, although recent articles have addressed some essential areas concerning the management of chronic wounds and their prevention [6]. The treatment of chronic wounds is therefore influenced, to varying degrees, by the practices of a workplace or physician. The mainstay of treatment follows the TIME principle: tissue debridement, infection control, moisture balance, and the edges of the wound. After these general measures have been addressed, the treatment is specific to the type of ulcer [5].

The development of chronic wounds is influenced by certain risk factors. The causes of impaired wound healing are often multifactorial. Local and systemic factors can contribute to the appearance of a wound and the length of wound resolution. Local factors (directly in the wound area) include a lack of oxygen supply to the skin (disturbance of blood circulation) and wound infections [7]. Bacteria rapidly colonize open skin wounds following compromised skin integrity. Microorganisms that colonize wounds typically involve the patient’s normal flora or may be transferred through any external contact with contaminated objects [3]. The presence of microbiological agents in a wound may not always be manifested by clinical signs of inflammation. In certain cases, the presence of bacteria alone is sufficient enough to slow down or completely halt the healing process. In such cases, we refer to it as a biofilm [8]. Systemic factors (concerning the whole organism) include being overweight, smoking, malnutrition, impaired mobility, diabetes, and sex (estrogens act protectively) [7]. A knowledge of risk factors and their consideration, especially at the beginning of the treatment of chronic wounds, brings with it better treatment results, including the prevention of complications and the reduction of treatment costs [9,10].

The research question explored whether certain anamnestic, anthropometric, or clinical factors exceed others in their role as a negative prognostic factor for wound healing. This study aimed to identify risk factors affecting the outcome of surgical care (healing vs. non-healing) for chronic wounds, and the duration of care required for healing.

## 2. Materials and Methods

### 2.1. Study Population

This study adopted an observational ambispective design (14 retrospective, 135 prospective). The studied population consisted of patients who underwent the treatment of the non-healing wounds of various etiologies at the surgical outpatient clinic of the Olomouc Military Hospital and completed the treatment between August 2021 and September 2023. The treatment was generally initiated by a general practitioner or sought by the patients themselves. The inclusion criteria included the completion of specialized care at a surgical outpatient clinic during the specified period and the availability of the patient’s personal medical history. The exclusion criteria included non-compliance with the treatment plan or termination of treatment by the patient, as well as the presence of multiple wounds in a patient. Among the enrolled patients, there were no individuals with confirmed neoplastic lesions, as the clinic where the study was conducted does not specialize in oncological patients.

During the initial examination, the patient’s personal medical history (mainly cardiometabolic and oncological diseases) was obtained by interviewing the patient and using any available medical reports. Anthropometric data were then obtained through a basic physical examination. The physical status classification system according to the American Society of Anesthesiologists (ASA) was determined, and the patient’s mobility was evaluated. According to the wound etiology, the patients were divided into five groups: diabetic ulcer, pressure ulcer (mostly heel or sacral), vascular ulcer (venous or arterial), wounds caused by injury (post-traumatic), and wounds caused by previous surgery (surgical site infection (SSI)). The initial characteristics of the lesion were also recorded—location, size (maximum diameter), depth (through disposable meter), state of the base, secretion (exudate) from the wound, and the state of the surrounding area of the wound.

Wound healing was managed using a uniform protocol including disinfection of the wound area, surgical debridement, and antiseptic lavage (with Betaine and Polyhexanide)—treatment directed at promoting moist wound healing [5,11]. The treatment took place in an outpatient setting with regular check-ups at 3-day intervals. Patients presenting advanced signs of inflammation (purulent secretion, presence of necrosis, or tissue debris) at the beginning or during the course of treatment were swabbed and tested for microbial infection before any further surgical treatment of the wound. In the case of a positive culture result, an appropriate antibiotic therapy was initiated, taking into account the microbiological agent (initially systemic therapy with continued local therapy if necessary). Patients presenting clinical signs of systemic inflammatory reactions (arising from infection) were admitted to hospital and commenced on intravenous antibiotic therapy with consideration for necrotomy and lavage under general anesthesia. At the beginning of admission, the concentration of the C-reactive protein (CRP) and the blood count were performed for every patient. The outcomes of treatment were healing, amputation, or death due to the lesion (and its complications).

### 2.2. Laboratory Analysis

The blood count was analyzed on Sysmex XN 1000 (Sysmex Europe SE, Brno, Czech Republic) using fluorescence flow cytometry. The determination of C-reactive proteins was carried out on Architect c8000 (Abbott Laboratories, Chicago, IL, USA) immunoturbidimetrically.

The Levine technique was applied to obtain a swab culture [12]. Cultures were carried out according to the standard microbiology culture procedures. Wound swabs were inoculated on chocolate agar, blood agar, and MacConkey agar. Using an inoculating loop, each sample was streaked onto the upper one-fourth of an agar plate with parallel overlapping strokes. The loop was heated and allowed to cool. The plate was turned at a right angle. The overlapping of the previous streak was performed once and then repeated on half of the remaining area. The final procedure was repeated with both the heated and cooled loop. Plates were incubated overnight at 35 °C–37 °C in an incubator. After incubation for 16–20 h, plates were checked for bacterial growth. Bacterial colonies were selected based on differences in shape, size, and color. Selected colonies from each plate were sub-cultured and incubated overnight. Bacterial identification was the final step of microbiological testing.

### 2.3. Statistical Analysis

Statistical analyses were conducted with the IBM SPSS Statistics, version 22 (SPSS Inc., Chicago, IL, USA). All numerical variables were characterized by descriptive statistics. Data distribution was checked using the Shapiro–Wilk test. Given the dominating right-skewed data distribution, the Mann–Whitney test for continuous variables and the chi-squared test for parametric variables were used for pairwise comparisons between the subgroups. The treatment outcome served as the response variable in a univariate logistic regression analysis. Only characteristics significantly associated with the outcome in the pairwise comparisons were included as explanatory variables. The effect size for each risk factor was estimated using odds ratio, the significance was assessed using *p*-values. The correlations of selected variables were quantified with Spearman’s correlation coefficient (r), and the level of significance (*p*) was determined. A *p*-value below 0.05 indicated statistical significance.

## 3. Results

### 3.1. Characteristics of the Study Population

The study population consisted of 149 patients (57 females, 92 males) with a mean age of 64.4 years (standard deviation 18; median 66; minimum 18, maximum 92 years). The median body mass index (BMI) was 28 kg/m^2^, indicating an increased portion of overweight patients. The most common causes of wounds were diabetic ulcers in 46 patients (30.9%), post-traumatic lesions in 38 patients (25.5%), pressure ulcers and SSI in 22 patients each (14.8%), and vascular ulcers in 21 patients (14.1%) (Table 1).

Almost half of the study sample (66 patients, 44.3%) were diabetics, of which 25 had a history of diabetic foot syndrome. A total of 23 patients (15.4%) had a prior history of lower limb amputation. Similar to diabetes mellitus, almost half of the sample population included a history of ischemic heart disease (73 patients, 49%). Furthermore, 108 patients (72.5%) had a history of arterial hypertension, and 80 patients (53.7%) had a history of hypercholesterolemia. The sample included 29 smokers (19.5%), 11 ex-smokers (7.4%), and 20 immobile patients (13.4%).

### 3.2. Initial Wound Characteristics

Significantly, 68.9% of patients presented with a wound depth exceeding 1 cm, while a shallower wound depth was found in 31.1% of patients. A similar distribution was observed in the presence of an inflammatory (68.2%) or granular base (31.8%), as well as secretion (62.8%) or the absence of secretion (37.2%) from the wound. Additionally, 75 patients (50.3%) showed a wound depth exceeding 1 cm, an inflammatory base, and wound secretion. The lesion surrounding the wound was non-inflammatory in 62.8% of patients. On the contrary, inflammatory reactions in the surrounding area occurred in 37.2% of cases. The wounds in their maximum diameter measured an average of 7 cm (median 5 cm) initially. The most common wound location included the lower extremities (66%), followed by the trunk (20.4%) and upper extremities (10.9%).

A total of 110 wounds were microbiologically examined. In 103 specimens, a bacterial agent was identified, based on which systemic antibiotic therapy was started (70 patients were treated with intravenous therapy during hospital admission and 43 patients were treated with oral therapy as outpatients). Microbiological cultures identified the presence of two-hundred bacterial colonies, with two bacterial genera predominating at the wound site (Table 2). The most commonly cultivated genera were *Staphylococci*, *Proteus*, and *Streptococci*. Fungal infection was not detected in any wound.

### 3.3. The Course and Outcome of Healing

Patient admission was indicated in 70 patients (47%), of which 38 required extensive surgical treatment under general anesthesia. The median CRP concentration at the beginning of patient admission was 89.8 mg/L, and the leukocyte concentration was 11 × 10^9^/L. In the entire sample, successful wound healing (with complete reepithelialization) occurred in 115 patients (77.2%), amputation in 21 (14.1%), and death due to wound complications in 13 patients (8.7%). Among the deceased patients, seven (53.8%) suffered from pressure ulcers, four (30.8%) from vascular, and two (15.4%) from diabetic foot ulcers. Diabetic ulcers as a cause of amputations was present in 10 patients (47.6%). The mean duration of healing was 110.9 days, and the median duration was 68 days (with a minimum 29 days and maximum 961 days).

### 3.4. Prediction of the Outcome of Healing

In terms of wound origin, the highest proportion of healed lesions was recorded in SSI (22; 100%) and post-traumatic ulcers (36; 94.7%). Among diabetic ulcers, complete healing occurred in 34 patients (73.9%), among vascular ulcers in 11 patients (52.4%) and pressure ulcers in 11 patients (50%). From the initial local characteristics, the wound depth, base, and secretion were significantly reflected in the distribution of the healing outcome. Among healed wounds, deeper wounds (>1 cm) were observed in 62.6% of cases, whereas, in non-healed wounds, greater wound depth was present in 88.2% (*p* = 0.006). An inflammatory base was present in 63.5% of healing wounds and 82.4% of non-healing wounds (*p* = 0.044). Wound secretion was initially present in 57.4% of healing wounds and 79.4% of non-healing wounds (*p* = 0.023). Healing wounds, when compared to non-healing wounds, initially presented a statistically significantly smaller maximum wound diameter (median 4 cm vs. 6.5 cm, *p* = 0.032). Table 3 illustrates the strength of the association between risk factors and healing outcomes in terms of odds ratios for statistically significant predictors.

Several statistically significant systemic risk factors were identified. The presence of all monitored diseases in a patient’s personal medical history was significantly higher in patients with non-healing wounds. The most significant difference was observed in the case of ischemic heart disease, which was present in 36.5% of patients with healing wounds compared to 91.2% in patients with non-healing wounds (*p* < 0.001). Patients with healing wounds were significantly younger compared to patients with non-healing wounds (a mean of 61 years vs. 76.1 years, *p* < 0.001). Patients with healing wounds were significantly less frequently on systemic antibiotic therapy during the treatment (66.1% vs. 94.1%, *p* = 0.003), less frequently admitted to hospital (46.1% vs. 73.5%, *p* = 0.006), scored significantly lower ASA scores (ASA 1 in 10.4% of healed vs. 0% of non-healed, ASA 4 in 18.3% of healed vs. 50% of non-healed, *p* < 0.001), underwent major surgical procedures under general anesthesia less often (37.4% vs. 67.6%, *p* = 0.003), were less frequently immobile (7.8% vs. 32.4%, *p* = 0.001), and less often administered long-term medication (76.7% vs. 97.1%, *p* = 0.012). In the representation of other monitored variables including sex, BMI, and cigarette smoking, both subgroups showed no significant differences. The effect sizes of the significant predictors of healing resulting from pairwise comparisons are shown in Table 3.

Although microbiological swabs were performed upon presentation of advanced signs of inflammation, healed and unhealed defects did not differ significantly in the representation of positive examinations (67% vs. 76.5%, *p* = 0.657), nor in the number of detected bacterial genera (*p* = 0.652), as seen in Table 2. Similarly, in terms of specific detected bacterial genera, no significant differences were found between healed and unhealed wounds (Table 4).

### 3.5. Determinants of the Healing Duration

The time from the initiation of wound treatment to successful healing significantly correlated with wound size (r = 0.247, *p* = 0.009) and the number of identified genera in the swab (r = 0.306, *p* = 0.006), and inversely correlated with the initial concentration of leukocytes in admitted patients (r = −0.349, *p* = 0.034). Among qualitative variables, significant determinants of prolonged healing included a wound depth greater than 1 cm (*p* < 0.001), an inflammatory base (*p* < 0.001), wound secretion (*p* = 0.001), and the presence of the genera *Proteus* (*p* = 0.02), *Pseudomonas* (*p* = 0.045), and *Escherichia* (*p* = 0.033).

Wounds requiring systemic antibiotic therapy (*p* < 0.001) and major surgical procedures (*p* = 0.006) resulted in longer healing times. The most significant variation in healing duration, as expressed by the median number of days, stemmed from the presence of *Proteus* (87 days) and *Pseudomonas* (72 days) bacterial genera, followed by wound secretion (56.5 days), the presence of an inflammatory base (56 days), a wound depth greater than 1 cm (55 days), and *Escherichia coli* infection (35 days).

## 4. Discussion

Multiple risk factors influencing secondary wound healing are well-documented, although the strength of association between specific factors and wound healing is variable. In this study, the overall portion of wounds healed was 77.2%. Significantly, 14.1% of patients underwent amputation, and wound-related mortality was 8.7%. Patients with chronic wounds exhibit elevated mortality rates, most notably in the case of diabetic foot ulcers, with reported five-year mortality rates ranging from 25% to 31% in some studies [3,13]. In the present study, the highest mortality was observed in individuals suffering from pressure ulcers (53.8%), compared to diabetic ulceration, which accounted for only 4.3% of deaths. Conversely, diabetic ulceration was the leading cause of amputation, accounting for 47.6% of cases.

The most common chronic wounds include venous and arterial leg ulcers, diabetic foot ulcers, and pressure ulcers, with occurrence reported depending on a specific population [4]. In our sample, diabetic ulcers (30.9%) prevailed, followed by pressure and vascular ulcers. For a comprehensive evaluation of the factors affecting wound healing, all types of skin lesions treated by the clinic were included in the study—covering post-trauma wounds and SSI (represented in a total of 40.8% of cases). The sample consisted of patients who were under the care of a hospital chronic wound center during a certain period, and the distribution of the wound origin was determined by the region- and content-specific organization of this type of healthcare.

Patients with chronic wounds require a multifactorial approach by healthcare providers, often necessitating interdisciplinary collaboration [13]. Only through an accurate assessment of the patient’s condition, wound status, and the identification of significant risk factors can we potentially minimize the treatment duration to the necessary minimum, consequently reducing overall therapy costs for the patient [10]. A thoroughly obtained medical history, in conjunction with records from the primary care physician or specialist, serves as a valuable source of patient data that can highlight patient risk factors and contribute towards the modification of the healing process, including the intensification of therapy if necessary.

In terms of patient history, the most significant factors influencing healing outcomes in our cohort appeared to be the presence of ischemic heart disease or other verified atherosclerotic diseases, along with associated conditions such as arterial hypertension, hypercholesterolemia, and diabetes mellitus, including diabetic foot syndrome. Diabetic and vascular wounds necessitate specific wound management in order to establish favorable conditions for healing [14,15]. Many chronic wounds fail to heal due to the stenosis or occlusion of the feeder arteries supplying the affected local tissue, termed peripheral arterial disease (PAD). Approximately 50% of patients with diabetic foot ulcers also exhibit PAD, underscoring the importance of vascular screening in these cases [4]. Diabetes plays a detrimental role in wound healing. It does so by affecting the healing process at multiple levels. Wound hypoxia, through a combination of impaired angiogenesis, inadequate tissue perfusion, and pressure-related ischemia are all major drivers of chronic diabetic wounds [16]. Chronic wounds are driven by metabolic disruptions, vascular deficits, or mechanical impact. In the absence of the available medical reports of the patient, it is advisable to carry out a basic screening for cardiovascular and metabolic diseases at the beginning of chronic wound care. Regarding the local wound characteristics, factors significantly impacting the healing outcome include an increased wound depth and maximum size of the wound, the presence of wound secretion, and an inflammatory base. Wound secretion is of particular concern, as excessive discharge affects patient comfort and requires frequent dressings [17]. Specialists should evaluate and consider these parameters when selecting appropriate wound-healing medications, as various products are often indicated for specific wound types [5,18]

The term chronic wound refers to wounds that have not healed within three months [19]. Considering this definition in our dataset, chronic wounds accounted for only 57 cases (38.3%), despite the average treatment duration exceeding three months, as indicated by the right-skewed data distribution. The shorter overall healing duration in the current study population may be due to the inclusion of post-traumatic wounds and SSI, which tend to heal earlier than wounds from diabetic ulcers [20].

The possible causes that transform a simple wound into a chronic wound are the object of study, and research has focused on infection as one of the crucial factors in developing and maintaining chronic wounds [19]. K. Jung et al. introduced a model that predicts the risk of slow-healing wounds [21]. The current study focused on the results of microbiological swabs, a paraclinical examination that plays an irreplaceable role in clinical practice. Knowledge of the specific etiological agent provides an opportunity for targeted antibiotic therapy, which may reduce the unnecessary use of systemic antibiotics in some patients [22]. Systemic antibiotic therapy is recommended for extensive defects that may potentially lead to sepsis [23]. Conversely, local antibiotic therapy offers the advantage of higher antibiotic doses, which, if administered systemically, could be toxic to the body [24]. Specific and targeted antibiotic therapy may not only improve wound conditions, but also serves a role in life-threatening infections. In our dataset, a total of 200 bacterial contaminations were detected in 103 positive samples. The specific microbial agents did not significantly predict differences between wound healing and non-healing.

In a non-negligible proportion of wound swab cultures, the results are related to microbial colonization (rather than infection), which may not affect the healing outcome [8]. However, the colonization of wounds by the bacteria of the genus *Proteus* extended the healing duration by 87 days, Pseudomonas by 72 days, and the presence of *E. coli* by 35 days. The *Proteus* genus is significantly linked to the stagnated/worsened evolution of wounds [25]. *Pseudomonas* genus is a group of organisms that are often detected in wounds; these organisms tend not to invade deeper tissues but can cause significant wound deterioration due to the production of tissue-destroying enzymes, antiphagocytic and adherence mechanisms, and exo- and endotoxins [8].

In other words, the presence of specific bacterial strains was not associated with specific outcomes of healing, but rather the duration of wound healing. This can also be explained by the fact that the previously mentioned bacterial genera belong to multidrug-resistant groups of bacteria, significantly complicating therapies across medical specializations, not just in the treatment of chronic wounds [26,27]. Aerobic or facultative pathogens, such as Staphylococcus, are commonly isolated from infected and clinically non-infected wounds. These organisms are relatively easy to cultures contributing to their frequent detection [8]. Staphylococcus was the most frequent genus revealed in the present study.

Swab culture is the most commonly used technique in the clinic due to its practical, noninvasive, reproducible, and cost-effective features. It has been reported that swab culture has a sufficient correlation with tissue biopsy to identify causative organisms in an infected wound. The major concern associated with swab culture is that only the superficial bacteria are reflected in the results, and deeper invasion by other pathogenic species may be missed. Moreover, swab cultures can be unreliable in the context of biofilm infection [28].

The limitations of this study involve the inclusion of the wounds of various etiologies, not exclusively those commonly considered as causes of chronic wounds. Standardized wound healing scales were not applied, which may limit comparisons with other studies [29]. The Levine swab technique may also represent a limitation, as it may not capture the microbial agent of deeper tissue layers. The possibility of patients being admitted to hospital due to occasional increases in inpatient bed availability is another limitation of this study. Hospital admission increases the risk of wound contamination by multidrug-resistant nosocomial bacterial species. On the other hand, it ensures a strict treatment regimen for the patient, thereby eliminating the possibility of incorrectly performed dressing changes in the home setting.

## 5. Conclusions

Based on a sample of the Czech population from a single surgical center focused on non-healing wounds, significant local and systemic risk factors affecting the healing process and determinants of healing duration were identified. Local factors critical to healing include wound depth, the presence of wound secretion, an inflammatory base, and wound size. Systemic factors that pose a risk to the development of chronic wounds include advancing age, immobility, and clinically manifesting atherosclerosis and its risk factors. The healing duration was most significantly influenced by the presence of *Proteus* and *Pseudomonas* bacterial genera. The early identification of local and systemic risk factors for chronic wounds can significantly contribute to successful healing, including a reduction in the duration of healing. The effective management of chronic wounds is pivotal for achieving better care outcomes and alleviating the risk of long-term disability.

## Figures and Tables

**Table 1 jcm-13-01003-t001:** Basic characteristics of the study population.

Characteristics	N	%
Number of patients	149	100
Male	92	38.3
Female	57	61.7
Medical history		
Arterial hypertension	108	72.5
Hypercholesterolemia	80	53.7
Ischemic heart disease	73	49
Diabetes mellitus	66	44.3
Immobility	20	13.4
Wound origin		
Diabetic ulcer	46	30.9
Post-traumatic lesions	38	25.5
Pressure ulcers	22	14.8
Surgical site infection	22	14.8
Vascular ulcers	21	14.1

**Table 2 jcm-13-01003-t002:** Results of microbiological examinations from 110 performed swabs.

Parameter	Specification	Result (*n*)
Result of a swab culture	Positive	103
	Negative	7
Bacterial genera in all swabs	*Staphylococcus*	55
	*Proteus*	28
	*Streptococcus*	26
	*Anaerobes*	24
	*Escherichia*	20
	*Enterobacter*	18
	*Pseudomonas*	12
	*Klebsiella*	10
	*Corynebacterium*	5
	*Citrobacter*	2
Number of detected genera in one swab (entire sample)	0	7
	1	36
	2	44
	3	17
	4	5
	5	1
Number of detected genera in one swab (healed wounds)	0	6
	1	28
	2	31
	3	11
	4	5
	5	1
Number of detected genera in one swab (unhealed wounds)	0	1
	1	8
	2	13
	3	6
	4	0
	5	0

**Table 3 jcm-13-01003-t003:** Risk factors significantly associated with amputation/death due to wound.

Risk Factors	Odds Ratio	Limits of 95% Confidence Interval		*p*-Value
		Lower	Upper	
Local factors				
Depth > 1 cm	4.375	1.441	13.282	0.009
Wound secretion	2.805	1.128	6.974	0.026
Inflammatory base	2.621	1.002	6.853	0.049
Wound size—largest diameter (cm)	1.080	1.022	1.141	0.006
Systemic factors				
Arterial hypertension	15.58	2.038	119.1	0.008
Ischemic heart diseases (clinically manifested atherosclerosis)	15.39	4.375	54.2	<0.001
Long-term medication	9.114	1.180	70.4	0.034
Hypercholesterolemia	8.378	2.727	25.7	<0.001
Systemic antibiotic treatment during healing	7.352	1.660	32.6	0.009
Immobility	5.580	2.074	15.0	0.001
ASA score	3.690	2.007	6.786	<0.001
Major surgical procedures under general anesthesia	3.477	1.506	8.025	0.004
Diabetic foot syndrome	2.937	1.287	6.700	0.010
Diabetes mellitus	2.494	1.094	5.686	0.030
Age (years)	1.063	1.032	1.095	<0.001

ASA, American Society of Anesthesiologists.

**Table 4 jcm-13-01003-t004:** Detection of bacteria in relation to the outcome of therapy.

Bacterial Genus	Outcome of Healing				*p*-Value
	Healed		Amputation/Death		
	N	% (Out of 82 Positive Smears)	N	% (Out of 28 Positive Smears)	
*Anaerobes*	16	19.5%	8	28.6%	0.316
*Citrobacter*	2	2.4%	0	0.0%	1.000
*Corynebacterium*	4	4.9%	1	3.6%	1.000
*Enterobacter*	13	15.9%	5	17.9%	0.774
*Escherichia (coli)*	17	20.7%	3	10.7%	0.235
*Klebsiella*	6	7.3%	4	14.3%	0.272
*Proteus*	18	22.0%	10	35.7%	0.149
*Pseudomonas*	8	9.8%	4	14.3%	0.497
*Staphylococci*	41	50.0%	14	50.0%	1.000
*Streptococci*	23	28.0%	3	10.7%	0.062

## Data Availability

The data presented in this study are available on request from the corresponding author.

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
