# Peer review of "Risk Factors for Non-Healing Wounds—A Single-Centre Study"

_jcm, 2024, doi:10.3390/jcm13041003_

Round 1

Reviewer 1 Report

Comments and Suggestions for Authors

Congratulations to the authors for quite an interesting study conducted. This topic is important, and the data collected in this study is also great. I have no doubts that the study is valuable - it has scientific and educational value.

The main question and the aim of the study is to identify the main factors that hinder proper wound healing process.

The topic is original. Chronic wounds account for 2-3% of the treated patients in the world.

The topic discussed is certainly addressed to specialists dealing with wounds treatment.

Firstly, he paper summarizes the results of other studies known so far, and secondly, it complements them with detailed research in this field.

The paper is well prepared in terms of methodology.

The conclusions respond to the stated research goal.

References are selected appropriately, only two or three position are over 15 years old.

The presented tables do not require additions.

It is important that the authors emphasize the need to develop standardized approach to the healing of individual cases. 

Author Response

Dear Reviewer 1,

Thank you very much for your review. We appreciate your time and valuable comments. We have prioritized our trying to comply with all reviewers' suggestions. Changes to the manuscript are highlighted in a track-changes system of Word. The manuscript has been corrected by a native speaker and a graphical abstract has been provided.

Reviewer´s comments:

Congratulations to the authors for quite an interesting study conducted. This topic is important, and the data collected in this study is also great. I have no doubts that the study is valuable - it has scientific and educational value.

The main question and the aim of the study is to identify the main factors that hinder proper wound healing process.

The topic is original. Chronic wounds account for 2-3% of the treated patients in the world.

The topic discussed is certainly addressed to specialists dealing with wounds treatment.

Firstly, he paper summarizes the results of other studies known so far, and secondly, it complements them with detailed research in this field.

The paper is well prepared in terms of methodology.

The conclusions respond to the stated research goal.

References are selected appropriately, only two or three position are over 15 years old.

The presented tables do not require additions.

It is important that the authors emphasize the need to develop standardized approach to the healing of individual cases. 

Reviewer 2 Report

Comments and Suggestions for Authors

Mauscript title: Risk Factors for Non-Healing Wounds - A Pilot Study from a 2 Single Centre

Authors:  

 Daniel Wolny  et. al

Comments to the Authors:

(A)  Provide an overview/summary of the manuscript

            This paper is a study with an observational ambispective design aimed to objectify the healing duration of chronic wounds of various etiologies and risk factors that can affect the course of healing 

            The topic of this article is of maxim interest because chronic wounds present a significant clinical, social, and economic challenge with a tremendous negative impact on patients’ quality of life and self-esteem. Few studies have examined the risk factors for the outcome of surgical care (healing vs. non-healing) for chronic wounds and the duration of care required for healing. Therefore, it is of utmost importance to study the factors that have an influence on treatment adherence in order to develop and implement appropriate management for as many patients as possible. A profound understanding of these factors allows us to identify obstacles and find solutions to enhance treatment adherence. Effective disease management goes beyond prescribing medications or procedures; it also involves psychological, social, and behavioral aspects.

            Thus, this article is extremely valuable and has great significance for the specialists who take care of such patients. 

            The manuscript is structured in four parts:

1.       Introduction

2.       Materials and Methods

3.       Results

4.       Discussion

(B) Introduction

            Daniel Wolny  et al. provided a comprehensive overview of chronic wounds, covering its clinical presentation, the impact that this skin condition has on patients’ quality of life and various treatment options. Moreover, the authors highlighted the psychological and social impact of the disease, such as lowering patients’ self-esteem and the importance of a multidisciplinary approach in the management of this condition. Furthermore, the differentiation in the vastness of the field of surgery, it may seem that surgical treatment of wounds is only a kind of marginal component. However given the ageing population, the increasing burden of diabetes and obesity worldwide, and the persistent problem of infection, it is expected that chronic wounds will continue to be a substantial clinical, social, and economic challenge.

            The aim of this study is also mentioned, which is to identify risk factors for the outcome of surgical care (healing vs. non-67 healing) for chronic wounds and the duration of care required for healing.

            The introduction is well documented and sets the context for the further information.

(C) Materials and Methods

            Materials and Methods are accurately described. The number of subjects involved in the study is mentioned along with inclusion criteria (149 patients (57 females, 92 males) with a mean 139 age of 64.4 (median 66) years) and exclusion criteria. The demographic and clinical information obtained from patients were also presented.

(D) Results

            The results are concise and are supplemented with three key tables.

            The quality of the data presented is good and the results appear to be valid.   The results reflect the methods in organization and structure.

(E) Discussion

            The Discussion section supports the data being presented. However, the authors did overreach and strayed away from their data specifically, plenty of other studies being mentioned, in the Discussion section.            

            In the last paragraph, the authors clearly stated what they have identified in their research, respectively: “Based on a sample of the Czech population from a single surgical centre focused on 317 non-healing wounds, significant local and systemic healing process risk factors and deter-318 minants of healing duration were identified. Early identification of local and systemic risk factors for chronic wounds can sig-324 nificantly contribute to successful healing, including reducing the healing duration. Effec-325 tive management of chronic wounds is pivotal for achieving better care outcomes and 326 alleviating adverse disabling consequences

(F) Quality of English language

            English is fine.

Author Response

Dear Reviewer 2,

Thank you very much for your detailed review. We appreciate your time and valuable comments. We have prioritized our trying to comply with all your suggestions. Changes to the manuscript are highlighted in a track-changes system of Word. The manuscript has been corrected by a native speaker and a graphical abstract has been provided.

Your comments (in Italics) have been addressed in the following way:

Mauscript title: Risk Factors for Non-Healing Wounds - A Pilot Study from a 2 Single Centre

Authors:  

 Daniel Wolny  et. al

Comments to the Authors:

(A)  Provide an overview/summary of the manuscript

            This paper is a study with an observational ambispective design aimed to objectify the healing duration of chronic wounds of various etiologies and risk factors that can affect the course of healing

             The topic of this article is of maxim interest because chronic wounds present a significant clinical, social, and economic challenge with a tremendous negative impact on patients’ quality of life and self-esteem. Few studies have examined the risk factors for the outcome of surgical care (healing vs. non-healing) for chronic wounds and the duration of care required for healing. Therefore, it is of utmost importance to study the factors that have an influence on treatment adherence in order to develop and implement appropriate management for as many patients as possible. A profound understanding of these factors allows us to identify obstacles and find solutions to enhance treatment adherence. Effective disease management goes beyond prescribing medications or procedures; it also involves psychological, social, and behavioral aspects.

            Thus, this article is extremely valuable and has great significance for the specialists who take care of such patients.

             The manuscript is structured in four parts:

  1. Introduction
  2. Materials and Methods
  3. Results
  4. Discussion

 (B) Introduction

            Daniel Wolny  et al. provided a comprehensive overview of chronic wounds, covering its clinical presentation, the impact that this skin condition has on patients’ quality of life and various treatment options. Moreover, the authors highlighted the psychological and social impact of the disease, such as lowering patients’ self-esteem and the importance of a multidisciplinary approach in the management of this condition. Furthermore, the differentiation in the vastness of the field of surgery, it may seem that surgical treatment of wounds is only a kind of marginal component. However given the ageing population, the increasing burden of diabetes and obesity worldwide, and the persistent problem of infection, it is expected that chronic wounds will continue to be a substantial clinical, social, and economic challenge.

            The aim of this study is also mentioned, which is to identify risk factors for the outcome of surgical care (healing vs. non-67 healing) for chronic wounds and the duration of care required for healing.

            The introduction is well documented and sets the context for the further information.

- The Introduction has been slightly expanded for a wider context.

(C) Materials and Methods

            Materials and Methods are accurately described. The number of subjects involved in the study is mentioned along with inclusion criteria (149 patients (57 females, 92 males) with a mean 139 age of 64.4 (median 66) yearsand exclusion criteria. The demographic and clinical information obtained from patients were also presented.

- Thank you for this point.

 (D) Results

            The results are concise and are supplemented with three key tables.

            The quality of the data presented is good and the results appear to be valid.   The results reflect the methods in organization and structure.

(E) Discussion

            The Discussion section supports the data being presented. However, the authors did overreach and strayed away from their data specifically, plenty of other studies being mentioned, in the Discussion section.

            In the last paragraph, the authors clearly stated what they have identified in their research, respectively: “Based on a sample of the Czech population from a single surgical centre focused on non-healing wounds, significant local and systemic healing process risk factors and determinants of healing duration were identified. Early identification of local and systemic risk factors for chronic wounds can significantly contribute to successful healing, including reducing the healing duration. Effective management of chronic wounds is pivotal for achieving better care outcomes and alleviating adverse disabling consequences ”

- Thank you for this point.

(F) Quality of English language

            English is fine.

Thank you again for your time!

Reviewer 3 Report

Comments and Suggestions for Authors

Authors shared a single centre experience of chronic wound healing management in an outpatient clinic. 
Authors aimed to determine the risk factors for prolonged wound healing in their cohort. 

Introduction - prolonged and provides excessive truisms about wound healing. IMO, in a clinical paper it may be written in more concise manner, focusing on local (country/state -wide) divergencies in wound healing management and problems 

M&M - few several flaws are distinctive. Study was mentioned to be "ambispective" - it is a cumbersome for further statistical analysis and drawing conclusions because of different settings of either retrospective or prospective studies. inclusion and exclusion criteria should be visibly described, along with time of observation (at least mean +/- SD). 
Standarized wound healing scales could have been incorporated (10.5216/ree.v20.49425 see table 2) 
  Number of prospective and retrospective cases should be presented. 
What statistical test was used to determine the OR, and what kind of comparison was made ? non amputated vs amputated/dead in the cohort for a particular factor confirmed with Chi-square? If so, it is a very low quality of evidence for proving the significance of single factors influencing the wound healing - due to mentioned multicomponent etiology (at one time) a logistic regression model should be use to determine the truly significant factors.
Results - population characteristics should be in a table. Further changes that should be made rely on proposed changes in the methodology. Tables are laconically described.

Discussion - prolonged and very general. It should focus on main findings of your study and confront them with he results from other centers in the country (even historic if available) or similar countries or general, recent findings from other similar centers. Microbiological findings were quite predictable (proteus and pseudomonas as main culprits of prolonged wound healing),yet there should be discussion on deciphering if the prevalence of other strains are exceptional for your population and is there a way to address it?
Conclusions - ok, yet but they should be updated according to aforementioned suggestions. If it is a pilot study - why it is called pilot, and what is next to be done?
Overall, at last major revision should be made, because of methodological and statistical issues that decrease the strength of the findings. 

Comments on the Quality of English Language

Overall the quality of English language in the manuscript is acceptable as for non-native speakers. Some clarification has to be done according to term "inflammation" - In my opinion authors misused this word with "infection" numerous times in the manuscript.

Author Response

Dear Reviewer 3,

Thank you very much for your review. We appreciate your time and valuable comments. We have prioritized our trying to comply with all your suggestions. Changes to the manuscript are highlighted in a track-changes system of Word. The manuscript has been corrected by a native speaker and a graphical abstract has been provided.

Your comments (in Italics) have been addressed in the following way:

Authors shared a single centre experience of chronic wound healing management in an outpatient clinic. 
Authors aimed to determine the risk factors for prolonged wound healing in their cohort. 

Introduction - prolonged and provides excessive truisms about wound healing. IMO, in a clinical paper it may be written in more concise manner, focusing on local (country/state -wide) divergencies in wound healing management and problems 

  • We wanted to provide a broad background so that the article would be useful even for non-specialists. All information given relates to the results of the study. All information given relates to the results of the study.

M&M - few several flaws are distinctive. Study was mentioned to be "ambispective" - it is a cumbersome for further statistical analysis and drawing conclusions because of different settings of either retrospective or prospective studies.

  • This study is designed as an ambispective due to data collection. All patients enrolled in the study have been monitored by the main author since March 2021; however, at that time, several patients (14) had already been treated at the surgical outpatient clinic of the hospital. Data were collected identically for both retrospective and prospective cases without the option of recall bias in the case of retrospectively included individuals. In other words, settings were the same for all patients. An influence on the statistical analysis is not probable. Our effort was to present the data collection procedure as accurately/as closely as possible to the truth in the Materials and Methods.

Inclusion and exclusion criteria should be visibly described, along with time of observation (at least mean +/- SD). 

  • Both inclusion and exclusion criteria are listed in the Methods at the end of the first paragraph. Patients were observed only during wound treatment. They were sent to the ambulance due to the wound - healing began at that moment. They did not attend the clinic after they healed. Thus, the length of observation is identical to the length of healing/treatment.

Standarized wound healing scales could have been incorporated (10.5216/ree.v20.49425 see table 2) 

  • Thanks for the helpful comment. We are inspired by a follow-up study. This "pilot" study was carried out according to the mentioned procedure and its transfer with the use of the scales is now impossible. We state the non-use of the scales in the limitations of the work.

 Number of prospective and retrospective cases should be presented. 

  • It has been added to the first sentence of the Materials and Methods.

What statistical test was used to determine the OR, and what kind of comparison was made ? non amputated vs amputated/dead in the cohort for a particular factor confirmed with Chi-square? If so, it is a very low quality of evidence for proving the significance of single factors influencing the wound healing - due to mentioned multicomponent etiology (at one time) a logistic regression model should be use to determine the truly significant factors.

  • Thank you for this point. We have revised the paragraph 2.3. Pairwise comparisons for non-healed vs. healed were performed using Chi-square (for parametric variables) and Mann-Whitney test (for continuous variables). Based on pairwise comparisons, effect sizes were determined for statistically significantly different variables using logistic regression and odds ratios.

Results - population characteristics should be in a table. Further changes that should be made rely on proposed changes in the methodology. Tables are laconically described.

  • Table 1 showing basic population characteristics has been added.

Discussion - prolonged and very general. It should focus on main findings of your study and confront them with he results from other centers in the country (even historic if available) or similar countries or general, recent findings from other similar centers. Microbiological findings were quite predictable (proteus and pseudomonas as main culprits of prolonged wound healing),yet there should be discussion on deciphering if the prevalence of other strains are exceptional for your population and is there a way to address it?

  • We have conducted a search and from our country - no other similar studies have been revealed (even historic). In the discussion, our findings are confronted with other works in several places. We have added a new section further addressing the relationship of bacteria to wound healing with associations to our results.

Conclusions - ok, yet but they should be updated according to aforementioned suggestions. If it is a pilot study - why it is called pilot, and what is next to be done?
Overall, at last major revision should be made, because of methodological and statistical issues that decrease the strength of the findings. 

  • It is a pilot study that will be followed up with the application of more standardized procedures. Since statistically significant and clear results were obtained even on this sample, it should not be necessary, in our opinion, to call it "a pilot". Therefore we have removed this designation from the manuscript title.

Overall the quality of English language in the manuscript is acceptable as for non-native speakers. Some clarification has to be done according to term "inflammation" - In my opinion authors misused this word with "infection" numerous times in the manuscript.

  • The language has been corrected by a native speaker. We have checked the use of “inflammation” vs “infection” throughout the manuscript.

Thank you again for your time!

Reviewer 4 Report

Comments and Suggestions for Authors

In generally this is very interesting paper describing ambispective observational pilot study on patients treated for a non-healing wounds in the surgical out-patient settings.

I have few comments which help this be more suitable for publication.

page 2 line 55-56. As you wrote - infection is a one of local factors impair wound healing. We decribe infection as local on the beginning, later general infection named sepsis can develop and cause patient death. Here you should also add to infection factor which we call critical colonization or biofilm formation - presence of bacteria in the wound without clinical signs of infection, which can also stops wound healing and cause keeping wound usually in inflamation phase.

page 59-61. Please add pain as factor playing role in chronic wound management with Ref: Moscicka P., Cwajda-Bialasik J. Jawien A et all.Occurence and Severity of Pain with Venous Leg Ulcers: A 12-week Longitudinal Study. 2020 Journal of Clinical Medicine. 9:(11) DOI: 10.3390/jcm9113399

line 85. Some chronic wounds in out-patient surgical clinic are neoplastic ulceration. Pleasy clarify that you did not observe neoplastic wounds or you did, but werent taken into study analysis.

line 87-89. Did you use any special tools for wound assessment eg. Bates-Jensen Wound Assessment Tool? If yes, please include. Did you measure only maximum diameter of the wound or wound surface by using eg. digital planimetry or digital photography

line 90-92. What kind of antiseptics were used? Do you have any special recommendations for antiseptic use eg. Kramers? Did you use antiseptic as lavaseptic for wound cleansing eg. PHMB or drug for treatment as wet gauze swabs eg.octenidyne? Please clarify and add ref. SopataM., Kucharzewski M., Tomaszewska E. Antiseptic with modern wound dressings in the treatment of venous leg ulcers:clinical and microbiological aspects. 2016 Journal of Wound Care 25(8),p.419-426.

page 3. line 113 In situation with critical colonisation or biofilm formation for microbiological assessment wound biopsy should be performed as reccomended. The Levine technique is still ok, but should be noted in Limitations of the study.

Page 3. Results. line 139. The data concerning age should be presented as eg.: mean 64.4 -+SD 12.4 (mediana 66; min 24 - max 86).

Page 4. line 153 better to use term exudate instead of secretion. Eg. Heavy, moderate, no exudate.

line 161 instead of defects better term wounds

line 164 instead of genera better term species

Were anaerobes tested? Any findings?

Please consider better presentation of main bacteria strains divided as Gram + positive and Gram - negative.

page 8. line 309. Limitations of the study could be selected as independent Chapter/Part 5.

On the end of the study were any patients who didnt finish observation period/did not attend scheduled visit, or just disapeared without any notice? If yes how many?

Author Response

Dear Reviewer 4,

Thank you very much for your review. We appreciate your time and valuable comments. We have prioritized our trying to comply with all your suggestions. Changes to the manuscript are highlighted in a track-changes system of Word. The manuscript has been corrected by a native speaker and a graphical abstract has been provided.

Your comments (in Italics) have been addressed in the following way:

In generally this is very interesting paper describing ambispective observational pilot study on patients treated for a non-healing wounds in the surgical out-patient settings.

I have few comments which help this be more suitable for publication.

page 2 line 55-56. As you wrote - infection is a one of local factors impair wound healing. We decribe infection as local on the beginning, later general infection named sepsis can develop and cause patient death. Here you should also add to infection factor which we call critical colonization or biofilm formation - presence of bacteria in the wound without clinical signs of infection, which can also stops wound healing and cause keeping wound usually in inflamation phase.

  • Information addressing your point has been added.

page 59-61. Please add pain as factor playing role in chronic wound management with Ref: Moscicka P., Cwajda-Bialasik J. Jawien A et all.Occurence and Severity of Pain with Venous Leg Ulcers: A 12-week Longitudinal Study. 2020 Journal of Clinical Medicine. 9:(11) DOI: 10.3390/jcm9113399

  • Thank you for this point. We are aware that even pain and social aspects speak to the healing process. However, in our study, we evaluated the association of biological variables with chronic wound healing. Therefore, we would rather not open this big topic in the manuscript.

line 85. Some chronic wounds in out-patient surgical clinic are neoplastic ulceration. Pleasy clarify that you did not observe neoplastic wounds or you did, but werent taken into study analysis.

  • We have added information about this into the manuscript (to the end of the first paragraph of the Materials and Methods).

line 87-89. Did you use any special tools for wound assessment eg. Bates-Jensen Wound Assessment Tool? If yes, please include. Did you measure only maximum diameter of the wound or wound surface by using eg. digital planimetry or digital photography

  • We applied disposable meters and only the maximum diameter was included. We have stated this in the manuscript.

line 90-92. What kind of antiseptics were used? Do you have any special recommendations for antiseptic use eg. Kramers? Did you use antiseptic as lavaseptic for wound cleansing eg. PHMB or drug for treatment as wet gauze swabs eg.octenidyne? Please clarify and add ref. SopataM., Kucharzewski M., Tomaszewska E. Antiseptic with modern wound dressings in the treatment of venous leg ulcers:clinical and microbiological aspects. 2016 Journal of Wound Care 25(8),p.419-426.

  • Prontosan was used - we do not mention the commercial name in the text, but we have added the names of the active substances. Management of wound healing is already cited in the text.

page 3. line 113 In situation with critical colonisation or biofilm formation for microbiological assessment wound biopsy should be performed as reccomended. The Levine technique is still ok, but should be noted in Limitations of the study.

  • Information addressing this point has been added to the second part of the Discussion. The application of the Ledvine technique has been added to the limitations.

Page 3. Results. line 139. The data concerning age should be presented as eg.: mean 64.4 -+SD 12.4 (mediana 66; min 24 - max 86).

  • It has been adjusted.

Page 4. line 153 better to use term exudate instead of secretion. Eg. Heavy, moderate, no exudate.

  • The word “exudate” has been added beyond the first mention of “secretion” into the Materials and Methods to make it clear throughout the manuscript.

line 161 instead of defects better term wounds

  • It has been replaced.

line 164 instead of genera better term species

  • Thank you for this point. We have checked it but the term “genus/genera” should be used when using a one-word term for a bacterial family such as Staphylococcus. Species refer to using a two-word term such as Staphylococcus aureus.

Were anaerobes tested? Any findings?

  • Yes, as shown in Table 4. However, there was no significant difference in their presence between both treatment outcomes.

Please consider better presentation of main bacteria strains divided as Gram + positive and Gram - negative.

  • Thank you for this point. We have considered this but as the number of bacterial genera detected is not that high and because of the clinical base of the paper we do not think it would help clarify the results.

page 8. line 309. Limitations of the study could be selected as independent Chapter/Part 5.

  • In other papers in the journal, it is usually only the last paragraph of the Discussion.

On the end of the study were any patients who didnt finish observation period/did not attend scheduled visit, or just disapeared without any notice? If yes how many?

  • Yes, this was an exclusion criterion as stated in the manuscript. We do not have an exact number, but we estimate about 20 patients during the follow-up period.

Thank you again for your time!

Reviewer 5 Report

Comments and Suggestions for Authors

Thank you for submitting this manuscript

Your research question is “whether there are anamnestic, anthropometric or clinical factors that exceed others in their role as negative prognostic factor for wound healing”. It addresses a specific gap in the field. 

It collects studies on different chronic wounds but did not consider the radiated ulcers, the patients who are immunocompromised or on chemotherapy. Why did not you add other local and systemic factors known to affect wound healing like radiation ulcers, autoimmune diseases or the use of chemotherapeutic drugs.

How was the depth of the wound measured? The author should comment on how did they calculated the depth.

Please do not use in the discussion “in our study”, instead you can use in this current study.

The authors mentioned that it is a pilot study, and mentioned the limitations of the study at the end of the discussion section. The conclusion consistent with the evidence and argument presented and addresses the main question posed.

The references are appropriate. 

The tables are well reconstructed.

Comments on the Quality of English Language

Needs English language revision

Author Response

Dear Reviewer 5,

Thank you very much for your review. We appreciate your time and valuable comments. We have prioritized our trying to comply with all your suggestions. Changes to the manuscript are highlighted in a track-changes system of Word. The manuscript has been corrected by a native speaker and a graphical abstract has been provided.

Your comments (in Italics) have been addressed in the following way:

Thank you for submitting this manuscript

Your research question is “whether there are anamnestic, anthropometric or clinical factors that exceed others in their role as negative prognostic factor for wound healing”. It addresses a specific gap in the field. 

It collects studies on different chronic wounds but did not consider the radiated ulcers, the patients who are immunocompromised or on chemotherapy. Why did not you add other local and systemic factors known to affect wound healing like radiation ulcers, autoimmune diseases or the use of chemotherapeutic drugs.

  • Oncology patients are not treated at the hospital where the study was conducted. Therefore, even associated wounds could not be part of this study. We have added information about this to the Materials and Methods.

How was the depth of the wound measured? The author should comment on how did they calculated the depth.

  • The depth of the wound was measured with a disposable tape measure. We included it in the manuscript.

Please do not use in the discussion “in our study”, instead you can use in this current study.

  • It has been replaced as you suggested.

The authors mentioned that it is a pilot study, and mentioned the limitations of the study at the end of the discussion section. The conclusion consistent with the evidence and argument presented and addresses the main question posed.

  • It is a pilot study that will be followed up with the application of more standardized procedures. Since statistically significant and clear results were obtained even on this sample, it should not be necessary, in our opinion, to call it "a pilot". Therefore, and also based on other reviewers´ comments, we have removed this designation from the manuscript title.

The references are appropriate. 

The tables are well reconstructed.

Needs English language revision

  • The language has been corrected.

Thank you again for your time!